# Corporate Social Responsibility and Employee Safety: Evidence from Korea

**Ja Eun Koo [1] and Eun Sun Ki [2,\*]**

[1] Division of Business Administration, The University of Suwon, Gyeonggi 18323, Korea; jekoo@suwon.ac.kr
[2] Division of Business Administration & Accounting, Kangwon National University, Chuncheon 24341, Korea
\* Correspondence: eski@kangwon.ac.kr

**Abstract:** Employees are an integral part of a company's sustainable growth and they expect a safe working environment. Therefore, analyzing the factors that affect employee safety is important. In this context, we analyze the effect of corporate social responsibility investment on employee safety. Using Korean listed company data from 2012 to 2014, we regress corporate social responsibility scores on workplace injuries. The Ordinary Least Square (OLS) regression results show that higher corporate social responsibility scores are associated with fewer working days lost owing to workplace injuries. Moreover, while workplace injuries have a clear negative effect on firm value, corporate social responsibility activity significantly reduces this negative effect. Our findings imply that investment in corporate social responsibility can improve workplace safety and contribute to a company's sustainable growth.

**Keywords:** employee safety; corporate social responsibility; workplace injury; sustainability

## 1. Introduction

The National Safety Council (NSC) reports that U.S. employers paid approximately US$55.43 billion in 2019 in employee compensation costs due to workplace injuries [1]. The cost of workplace injuries is greater than the total cost of treating cancer patients [2]. Workplace injuries can erode a firm's profitability and undermine its ability to recruit and retain employees [3]; employees desire a safe workplace, so employee safety is indispensable for a company's sustainable growth. Workplace safety has been extensively studied in the field of management, but its relationship with corporate social responsibility (CSR) has not yet been explored. This study investigates whether socially responsible firms invest more in workplace safety, resulting in fewer working days lost due to workplace injuries. Moreover, we examine whether the negative effect of workplace injuries on firm value is reduced by CSR activities.

We argue that CSR is associated with employee safety in two main ways. First, workplace injuries have a negative impact on corporate reputation, reducing the effectiveness of CSR. In recent times, CSR has become a popular means of managing corporate reputation [4]. The literature suggests that CSR improves relationships with stakeholders and allows a firm to build a favorable reputation for itself, which improves the firm's ability to recruit and retain talented employees [5–7], enhances customer loyalty [8], strengthens supplier commitment [9], and increases investor confidence [10–12]. Firms with a focus on CSR seek to lower workplace injury rates to reduce the potential for damage to their corporate reputation. Second, employees are important internal stakeholders that companies should consider in CSR activities. Employees share common interests with a firm's success; therefore, meeting their expectations is critical to long-term sustainable growth [13]. Socially responsible firms recognize the negative impact of workplace injuries on employee morale and well-being [14] and will invest

accordingly in employee safety. Thus, we expect a positive relationship between a high level of CSR and employee safety.

We explore the effect of CSR on workplace safety using firm-level injury data from the Korea Occupational Safety and Health Agency, which operates under the aegis of the Ministry of Employment and Labor and collects workplace injury data and produces related statistics. We begin by examining the empirical relationship between firm-level injury data and CSR, using environmental, social, and governance scores to measure firm-level CSR activities. The scores are obtained from the Korea Corporate Governance Service, which provides a comprehensive measure of the sustainability of listed companies in terms of environmental responsibility, social responsibility, and governance.

Using Korean listed company data from 2012 to 2014 and controlling for firm characteristics and industry-year fixed effects, we find a negative relationship between working days lost due to workplace injuries and CSR scores. The empirical results support the hypothesis that socially responsible companies invest more in employee safety and have fewer workplace injuries. We then investigate the impact of CSR activities on the relationship between workplace injuries and firm value. Workplace injuries result in significant direct and indirect costs that can reduce the value of a firm. However, socially responsible companies negate this impact by being active in preventing and resolving the damage caused by workplace injuries. Our empirical results show that the number of working days lost due to workplace injuries is negatively related to firm value, but this negative relationship is weak in companies with high CSR scores.

Our study contributes to the literature in several ways. First, we expand the research on the determinants of employee safety to include CSR. Prior studies document that financial constraints adversely affect employee safety [15,16], but other factors have not been addressed. Second, our study explores the benefits of CSR from an employee perspective, rather than from a financial performance perspective. The literature emphasizes that CSR activities lead to an improvement in firm value or financial performance [8,10–12]; However, there is little research on how CSR activities benefit employees. Our study fills this gap. Third, whereas most prior literature employs establishment-level injury data [15], we analyze firm-level injury data, which will help us improve our understanding of the determinants of firm-level injuries.

Neither the United States nor Korea has disclosed firm-level injury data to the public. The Korean data used in this study is unique in that it provides information on the number of working days lost owing to workplace injuries at the individual firm level. Because the social costs of workplace accidents are of common interest not only in Korea, but also worldwide, policymakers in every country are working to reduce workplace injuries. For example, U.S. employers paid US$55.43 billion in 2019 in compensation costs for injured employees [1]. For government agencies that manage workplace accidents, it is important to select companies with a high probability of incidents and take precautionary measures. Our findings imply that policy makers need to consider individual firms' CSR investments when screening for companies where workplace accidents are highly probable.

This article is structured as follows. Section 2 reviews prior studies and establishes our hypotheses. Section 3 discusses the sample and research design, Section 4 presents the empirical results, and Section 5 summarizes and concludes the research.

## 2. Background and Hypotheses Development

### 2.1. Literature Review

Over the past two decades, the global emphasis on CSR has steadily increased. CSR is no longer considered to be a matter of choice; it is an integral part of corporate sustainable growth. While the definition of CSR is inconclusive, the most widely accepted definition is provided by Aguinis [17], who defines CSR as "context-specific organizational actions and policies that take into account stakeholders' expectations and the triple bottom line of economic, social, and environmental performance". CSR is costly because it considers both internal and external stakeholders. According to a survey of

1000 top-ranking companies in sales and public institutions, as of 2017, Korean companies spend an average of 0.14% of their earnings on CSR activities [18]. The emphasis on CSR has prompted researchers to determine the benefits to firms of investing in CSR activities. Most studies show that CSR improves the perceptions of external stakeholders, resulting in an improvement in financial performance. For example, customers that value the CSR activities of a firm will have a preference for the product or service the firm provides, leading to customer satisfaction, customer loyalty, and increased sales [8,19].

Socially responsible companies are less likely to engage in earnings management and provide more reliable financial information [20,21]. The literature shows that investors and creditors have more confidence in companies with high CSR scores, thus CSR provides the benefits of positive abnormal stock returns, high institutional ownership, high credit ratings, and low capital costs [10–12,22]. Therefore, several studies highlight that CSR is a necessary expenditure.

Product market competition can affect CSR activities. In developed countries, companies in competitive industries engage in more CSR activities [23,24]. However, in an emerging market, by contrast, companies in non-competitive industries engage in more CSR activities [25]. Executive compensation may also affect CSR activities. Monetary incentives designed to align the CEO's and shareholders' interests have a negative effect on CSR, whereas non-monetary incentives have a positive effect [26]. Meanwhile, CEO risk taking incentives increase firm risk only in low CSR firms and have no effect on firm risk in high CSR firms. High CSR firms attempt to balance the interests of investing and non-investing stakeholders, so CEO risk taking incentives have no effect on firm risk [27]. Recent research on CSR show that corporate visibility in print media has a positive significant relationship with CSR ratings [28,29].

Research on the relationship between CSR and its effect on employees is rare but has increased in recent years. Ramus and Steger [30] argue that employees are more actively involved in CSR activities if they have a positive attitude toward their firm's CSR strategy. CSR also improves a firm's ability to attract and retain talent. Job seekers lack information about the company to which they apply and infer companies with high CSR rankings as being fair and reputable [5,6]. In addition, employees of companies that participate actively in CSR have high levels of job satisfaction and organizational identification, resulting in a low turnover of personnel [7,31–34]. While prior research focuses on the impact of CSR on employee satisfaction or organizational identification, this study investigates the practical benefits employees can obtain from CSR activities, such as a safer working environment.

### 2.2. Hypotheses Development

### 2.2.1. CSR and Employee Safety

Socially responsible firms are incentivized to behave honestly, reliably, and ethically because such behaviors provide benefits [35]. The literature shows that a positive reputation gained through CSR contributes to increasing sales, attracting talent, and raising firm value [5–12]. In contrast, frequent workplace injuries can damage a firm's reputation. If a company receives negative media attention due to a workplace injury, it loses public trust, which can lead to customer churn, a decline in stock prices, increased capital costs, and a lower credit rating [36]. A key purpose of CSR is to build a favorable reputation by improving relationships with stakeholders; therefore, socially responsible firms will seek to reduce the number of workplace injuries.

Employees are important company stakeholders, and firms must work to understand and meet their expectations. ISO 26000, which provides guidance on corporate social responsibility, emphasizes that employees share a common interest with the company's purpose and its success [13], so meeting employees' expectations is important for long-term corporate growth. A safe workplace is important to employees; workplace accidents threaten mental and physical health, and socially responsible companies will, therefore, invest more in workplace safety to meet employee expectations of a safe work environment.

Workplace accidents can threaten the safety of the local community in which the workplace is located and can cause serious environmental problems [14]. The local community is an important stakeholder for companies to consider when engaging in CSR [13]. Therefore, companies that want to maintain a close relationship with the local community will make an active effort to prevent workplace injuries.

Based on this reasoning, we expect a negative relationship between CSR performance and workplace injuries; that is, a positive relationship between CSR performance and employee safety. To examine the impact of CSR performance on workplace injuries, we set the first hypothesis as follows:

**Hypothesis 1.** CSR is negatively associated with workplace injuries.

### 2.2.2. Employee Safety and Firm Value, and CSR

Frequent workplace injuries have a negative impact on the value of a firm due to the resulting direct and indirect costs. Direct costs include medical costs, compensation claim costs for injured employees, and damage to business property. Indirect costs include those arising from production downtime, lower productivity, lower employee morale, damage to the corporate reputation, and difficulty in attracting talented employees. Cohn and Wardlaw [37] find a negative relationship between firm value and workplace injury rates. They show that firm value decreases by 6.1% when the injury rate increases by one standard deviation.

The negative impact workplace injuries on firm value may be weakened if companies are active in injury prevention and resolve issues caused by workplace injuries by engaging with employees and the local community. To examine the effect of CSR on the relationship between workplace injuries and firm value, we set the second hypothesis as follows:

**Hypothesis 2.** CSR performance weakens the negative relationship between workplace injuries and firm value.

## 3. Methodology and Data

### 3.1. Methodology

Our first hypothesis examines whether CSR is a determinant of the rate of workplace injuries. To test this argument, we set the following Ordinary Least Square (OLS) regression model. Table 1 presents the definitions of the variables used in Equation (1).

$$\text{Ln(WDL)}_{it+1} = \alpha_0 + \alpha_1 \text{CSR}_{it} + \alpha_2 \text{SIZE}_{it} + \alpha_3 \text{AGE}_{it} + \alpha_4 \text{MCH}_{it} + \alpha_5 \text{CFO}_{it} + \alpha_6 \text{ROA}_{it} + \alpha_7 \text{LEV}_{it} + \alpha_8 \text{GROW}_{it} + \alpha_9 \text{WB}_{it} + \alpha_{10} \text{Ln(EMP)}_{it} + \text{Industry fixed effect} + \text{Year fixed effect} + \varepsilon \tag{1}$$

**Table 1.** Variable definition in Equation (1).

| Variables | | Definition |
|---|---|---|
| Ln(WDL) | = | the log of working days lost due to workplace injuries in year t + 1 |
| CSR | = | the firm's CSR performance in year t |
| SIZE | = | the log of the firm's total assets in year t |
| AGE | = | the firm's age in year t |
| MCH | = | the value of the firm's machinery divided by depreciable tangible assets in year t |
| CFO | = | the operating cash flow of the firm divided by total assets in year t |
| ROA | = | the firm's return on assets (net income/total assets) in year t |
| LEV | = | the firm's leverage ratio (total liabilities/total assets) in year t |
| GROW | = | the firm's sales growth rate in year t |
| WB | = | the value of welfare benefits per employee of the firm in year t |
| Ln(EMP) | = | the log of the number of employees in the firm in year t |
| $\varepsilon$ | = | error term |

The dependent variable, Ln(WDL), is the number of working days lost due to workplace injuries and reflects the severity of workplace injuries. Workplace injury rates treat both simple and death accidents in the same case, so the seriousness of the accidents is not reflected. On the other hand, the number of working days lost increases with the grade of the worker's physical disability, which has the advantage of reflecting the severity of the accident. For example, accidents with the smallest physical disability are calculated as 50 working days lost per case, while deaths are calculated as 7500 working days lost per case. The variable of interest in Equation (1) is CSR. If CSR reduces workplace injuries by improving workplace safety, $\alpha_1$ is expected to be negative. Note that in Equation (1), the dependent variable is the (t + 1) year value, while the explanatory variables are the t year values. This is because we assume that current investments in workplace safety will affect the workplace injury rate for the following year.

In accordance with Cohn and Wardlaw [16], we include firm size (SIZE), firm age (AGE), machinery ratio (MCH), operating cash flows (CFO), profitability (ROA), debt ratio (LEV), and sales growth rate (GROW) to control for the impact of firm characteristics on the rate of workplace injuries. Cohn and Wardlaw [16] argue that firms with financial constraints are less likely to invest in employee safety, resulting in a higher risk of injury. Firm size (SIZE), operating cash flows (CFO), and profitability (ROA) are inverse proxy variables for financial constraints. The impact of firm age (AGE) on the rate of workplace injuries is inconclusive. The risk of injury may be low because older companies are more efficient in safety management; or conversely, older companies may have more frequent accidents due to the deterioration of facilities. We control for machinery ratio (MCH) because workplace injuries occur more frequently in industries related to physical assets rather than services. We also control for sales growth rate (GROW); growing companies may lack the capacity to invest in employee safety because of the burden of reinvestment, and this lack of investment can increase the risk of injury. We include the number of employees, Ln(EMP), as a control variable because a company with a large number of employees will often have a higher number of working days lost. Finally, we include year and industry dummy variables to control for year- and industry-specific differences in workplace injuries.

Our second hypothesis examines whether CSR is a moderating variable in the relationship between workplace injuries and firm value. To test this argument, we set the following OLS regression model. Table 2 presents the definitions of the variables used in Equation (2).

$$\begin{aligned} TQ_{it} = {} & \beta_0 + \beta_1 WDLD_{it} + \beta_2 CSR_{it} + \beta_3 WDLD_{it} \times CSR_{it} + \beta_4 LEV_{it} + \beta_5 CASH_{it} + \beta_6 SIZE_{it} + \\ & \beta_7 TAR_{it} + \beta_8 CFO_{it} + \beta_9 DIV_{it} + \beta_{10} ATO_{it} + \text{Industry fixed effect} + \text{Year fixed effect} + \varepsilon \end{aligned} \quad (2)$$

**Table 2.** Variable definition in Equation (2).

| Variables | | Definition |
|---|---|---|
| TQ | = | Tobin's Q in year t |
| WDLD | = | 1 if a firm's working days lost due to workplace injuries in year t is greater than the median of the sample, or otherwise is 0 |
| CSR | = | the firm's CSR performance in year t |
| LEV | = | the firm's leverage ratio (total long-term liabilities/total assets) in year t |
| CASH | = | the firm's cash level divided by total assets in year t |
| SIZE | = | the log of the firm's total assets in year t |
| TAR | = | the firm's tangible asset ratio (total liabilities/total assets) in year t |
| CFO | = | the firm's operating cash flow divided by total assets in year t |
| DIV | = | the firm's dividends divided by total assets in year t |
| ATO | = | the firm's asset turnover ratio (sales/total assets) in year t |
| $\varepsilon$ | = | error term |

The dependent variable is Tobin's Q, which is calculated as the sum of the market value of equity and debt divided by total assets. In Equation (2), the variable of interest is Ln(WDL) × CSRD, which represents the interaction between workplace injuries and CSR performance. If workplace injuries have

a negative impact on firm value but this relationship is weaker in firms with a good CSR performance, we expect $\beta_1$ to be negative and $\beta_2$ to be positive. We also include firm-specific controls that affect firm value, in line with Cohn and Wardlaw [16].

*3.2. Data*

Our sample consists of non-financial companies listed on the Korea Stock Exchange for the 2012 to 2014 period. All companies in the sample have three key characteristics: they have workplace injury data and CSR scores, their financial data is available on TS2000, and their fiscal year ends in December. Financial firms are excluded due to their low comparability with other sectors. Our data on firm-level workplace injury is obtained from the Korea Occupational Safety and Health Agency, which is commissioned by the Ministry of Employment and Labor to collect workplace injury data and produce related statistics. Workplace injury data is only available for the 2012 to 2014 period; therefore, our study is limited to that period. CSR performance is measured by environmental, social, and governance scores provided by the Korea Corporate Governance Service. Since 2011, the Korea Corporate Governance Service has evaluated the level of sustainability management of Korean listed companies in the key CSR aspects of environmentally responsible management, social responsibility, and corporate governance. We also require the availability of financial data from TS2000. TS2000 is a business information service system that provides financial information in business reports and audit reports submitted online by Korean listed companies. To reduce distortion in the sample due to outliers, we winsorize all continuous variables at the top and bottom percentile. Based on these criteria, our final sample consist of 1234 firm-year observations. Table 3 shows the sample distribution by year. A similar distribution is shown for each year, ranging from 409 in 2012 to 417 in 2014.

**Table 3.** Sample Distribution by Year.

| Year. | Frequency | % |
|---|---|---|
| 2012 | 409 | 33.14 |
| 2013 | 408 | 33.06 |
| 2014 | 417 | 33.79 |
| Total | 1234 | 100.00 |

Table 4 presents the sample distribution by industry. The manufacturing industry represents the highest proportion of the sample at 72.93%, followed by wholesale and retail trade (8.18%) and professional, scientific, and technical activities (5.83%).

**Table 4.** Sample Distribution by Industry.

| Industry [1] | Frequency | % |
|---|---|---|
| Mining and quarrying | 3 | 0.24 |
| Manufacturing | 900 | 72.93 |
| Electricity, gas, steam and air conditioning supply | 23 | 1.86 |
| Construction | 68 | 5.51 |
| Wholesale and retail trade | 101 | 8.18 |
| Transportation and storage | 24 | 1.94 |
| Information and communication | 34 | 2.76 |
| Professional, scientific and technical activities | 72 | 5.83 |
| Business facilities management and business support services; rental and leasing activities | 6 | 0.49 |
| Education | 3 | 0.24 |
| Total | 1234 | 100.0 |

[1] We use a one-digit SIC code for industry classification.

## 4. Empirical Results

### 4.1. Descriptive Statistics

Table 5 presents the descriptive statistics for the sample. The average Ln(WDL) value is 3.9809, indicating that the average firm loses 53.6 working days per year due to workplace injuries. The mean and standard deviation for Ln(WDL) are 3.9809 and 3.7539, respectively, which implies that the severity of workplace injuries varies across the sample. The average CSR score is 5.7464, while the minimum and maximum values are 5.0626 and 6.5103, respectively. The average Tobin's Q is 0.5456, which means the market value of the total assets in our sample is 0.5456 times the book value, on average. The average Ln(Assets) value is 26.8480, indicating that the average firm has total assets of KRW57 billion. The companies in our sample have an average age of 40 years since their establishment. The average firm reports 1.01% of its total assets as earnings (ROA), and its average debt ratio (LEV) is 45.48%. On average, operating cash flow (CFO) and tangible assets (TAR) account for 4.94% and 31.22% of total assets, respectively. The average Ln(EMP) value is 6.6128, indicating that the average firm has 745 employees.

**Table 5.** Descriptive statistics (N = 1234).

| Variables [1] | Mean | Standard Deviation | Minimum | Maximum |
|---|---|---|---|---|
| $Ln(WDL)_{t+1}$ | 3.9809 | 3.7539 | 0.0000 | 11.2975 |
| $CSR_t$ | 5.7464 | 0.3082 | 5.0626 | 6.5103 |
| $TQ_t$ | 0.5456 | 0.1844 | 0.1275 | 0.9114 |
| $SIZE_t$ | 26.8480 | 1.5343 | 23.8807 | 31.4403 |
| $AGE_t$ | 40.0810 | 16.9995 | 4.0000 | 85.0000 |
| $MCH_t$ | 0.3359 | 0.2487 | 0.0000 | 0.8364 |
| $ROA_t$ | 0.0101 | 0.0730 | −0.3612 | 0.1564 |
| $CFO_t$ | 0.0494 | 0.0889 | −0.3046 | 0.3868 |
| $LEV_t$ | 0.4548 | 0.1854 | 0.0886 | 0.9225 |
| $WF_t$ | 0.1065 | 0.0594 | 0.0000 | 0.2965 |
| $Ln(EMP)_t$ | 6.6128 | 1.6712 | 3.3322 | 11.6634 |
| $CASH_t$ | 0.0000 | 0.0001 | $9.79e-08$ | 0.0003 |
| $TAR_t$ | 0.3122 | 0.1774 | 0.0018 | 0.7168 |
| $DIV_t$ | 0.0062 | −0.0078 | 0.0000 | 0.0430 |
| $ATO_t$ | 0.9479 | 0.4896 | 0.1006 | 2.8590 |

[1] Refer to Table 1; Table 2 for the definitions of variables.

Table 6 presents the Pearson correlations among the variables used in Equation (1). When the other variables are not controlled, the rate of workplace injuries is positively related to CSR performance in the previous year. These results are inconsistent with our expectations. In addition, the correlations indicate that the severity of workplace injuries is higher for companies with large total assets (SIZE), many employees (Ln(EMP)), old age (AGE), a high machinery ratio (MCH), and a high debt ratio (LEV). Meanwhile, CSR performance is positively related to firm size (SIZE), the number of employees (Ln(EMP)), machinery ratio (MCH), profitability (ROA), debt ratio (LEV), and welfare benefits per employee (WF).

Table 7 presents the Pearson correlations among the variables used in Equation (2). Tobin's Q is negatively related to working days lost due to workplace injuries, consistent with prior studies [16]. Moreover, Tobin's Q is negatively related with CSR, firm size (SIZE), debt ratio (LEV), tangible asset ratio (TAR), and asset turnover ratio (ATO), while it is positively related with cash holdings (CASH), operating cash flows (CFO), and dividend ratio (DIV).

We check for the variance inflation factor (VIF) in all regression models. The VIFs are between 1.06 and 4.74, indicating that multicollinearity is not a serious concern.

**Table 6.** Pearson correlations of the variables used in Equation (1).

| Variables | Ln(WDL) | CSR | SIZE | AGE | MCH | ROA | LEV | WF |
|---|---|---|---|---|---|---|---|---|
| CSR | 0.39 *** (0.00) | | | | | | | |
| SIZE | 0.51 *** (0.00) | 0.74 *** (0.00) | | | | | | |
| AGE | 0.07 ** (0.02) | 0.03 (0.30) | 0.08 ** (0.01) | | | | | |
| MCH | 0.19 *** (0.00) | 0.27 *** (0.00) | 0.16 *** (0.00) | 0.03 (0.33) | | | | |
| ROA | 0.04 (0.12) | 0.11 *** (0.00) | 0.16 *** (0.00) | −0.01 (0.61) | 0.12 *** (0.00) | | | |
| LEV | 0.21 *** (0.00) | 0.11 *** (0.00) | 0.15 *** (0.00) | −0.02 (0.39) | −0.07 ** (0.01) | −0.33 *** (0.00) | | |
| WF | 0.00 (0.94) | 0.18 *** (0.00) | 0.20 ***(0.00) | −0.02 (0.40) | −0.03 (0.26) | 0.01 (0.66) | −0.04 (0.19) | |
| Ln(EMP) | 0.62 *** (0.00) | 0.64 *** (0.00) | 0.79 *** (0.00) | 0.02 (0.41) | 0.00 (0.91) | 0.07 ** (0.01) | 0.27 *** (0.00) | 0.17 *** (0.00) |

Refer to Table 1 for the definitions of variables. P values are in parentheses. *, **, and *** represent statistical significance at the 10%, 5%, and 1% level, respectively.

**Table 7.** Pearson correlations of the variables used in Equation (2).

| Variables | TQ | Ln (WED) | CSR | CASH | LEV | SIZE | TAR | CFO | DIV |
|---|---|---|---|---|---|---|---|---|---|
| Ln (WED) | −0.18 *** (0.00) | | | | | | | | |
| CSR | −0.11 *** (0.00) | 0.31 *** (0.00) | | | | | | | |
| CASH | 0.17 *** (0.00) | −0.11 *** (0.00) | −0.05 * (0.07) | | | | | | |
| LEV | −0.50 *** (0.00) | 0.21 *** (0.00) | 0.32 *** (0.00) | −0.16 *** (0.00) | | | | | |
| SIZE | −0.15 *** (0.00) | 0.40 *** (0.00) | 0.74 *** (0.00) | −0.07 ** (0.01) | 0.15 *** (0.00) | | | | |
| TAR | −0.07 ** (0.01) | 0.14 *** (0.00) | 0.16 *** (0.00) | −0.25 *** (0.00) | 0.07 ** (0.01) | 0.03 (0.27) | | | |
| CFO | 0.28 *** (0.00) | 0.08 *** (0.00) | 0.12 *** (0.00) | 0.14 *** (0.00) | −0.28 *** (0.00) | 0.09 *** (0.00) | 0.16 *** (0.00) | | |
| DIV | 0.39 *** (0.00) | 0.03 (0.27) | 0.06 ** (0.02) | 0.11 *** (0.00) | −0.39 *** (0.00) | −0.02 (0.54) | 0.01 (0.63) | 0.37 *** (0.00) | |
| ATO | −0.17 *** (0.00) | 0.02 (0.57) | −0.03 (0.30) | 0.11 *** (0.00) | 0.17 *** (0.00) | −0.07 ** (0.01) | −0.06 ** (0.04) | 0.19 *** (0.00) | 0.10 *** (0.00) |

Refer to Table 2 for the definitions of variables. P values are in parentheses. *, **, and *** represent statistical significance at the 10%, 5%, and 1% level, respectively.

## 4.2. Regression Results

Table 8 shows the regression results for Equation (1). The dependent variable is the natural log of year (t + 1) working days lost owing to injuries. The coefficient on CSR is −1.581 and is statistically significant at the 1% level. This means that CSR investments in a given year result in a reduction in working days lost owing to injuries in the next year. Because socially responsible firms recognize the negative effect of workplace injuries on employee morale and well-being [14], they will invest more in employee safety. In turn, active employee safety investments by CSR firms will reduce working days lost in the following year. The negative relationship between workplace injuries in a given year and CSR performance in the prior year supports these arguments.

The regression results for the other control variables are as follows. Working days lost owing to injuries are positively related to both SIZE and Ln(EMP). Workplace accidents occur more frequently at larger companies than smaller ones. As a result, larger companies have more working days lost owing to injuries. However, this result is inconsistent with Cohn and Wardlaw [16]. They document a negative relationship between injury rates and firm size. Whereas they use establishment-level injury

rates as a dependent variable, we use firm-level working days lost owing to injuries. Because of these differences in research design, our findings may be greatly influenced by firm size. We replicate Table 8 using working days lost owing to injuries per employee [ = working days lost owing to injuries/the number of employees] as a dependent variable to more accurately control the effect of firm size on our results (See Tables 10 and 11). However, our findings do not change qualitatively. Meanwhile, the coefficient of MCH is positive and statistically significant at the 1% level. This indicates that workplace injuries are more relevant to industries that manufacture products using machinery than to service industries. This result is line with Cohn and Wardlaw [16]. The coefficient of LEV is also significantly positive at the 10% level and implies that companies with high debt ratios are financially constrained, so they invest less in employee safety, resulting in higher injury rates [16]. This result is also consistent with Cohn and Wardlaw [16]. Lastly, the coefficient on WB is significantly negative at the 1% level. This is in line with our predictions, implying that companies with low employee welfare spending are at greater risk for workplace accidents. Adj. $R^2$, which indicates the model's explanatory power, is relatively high at 44.0%.

**Table 8.** The relationship between corporate social responsibility (CSR) and workplace injuries.

| Variables [1] | Coefficients | | t-Value |
|---|---|---|---|
| CSR | −1.581 | *** | −3.629 |
| SIZE | 0.313 | *** | 2.756 |
| AGE | 0.007 | | 1.486 |
| MCH | 2.890 | *** | 7.419 |
| ROA | −0.644 | | −0.523 |
| LEV | 0.861 | * | 1.739 |
| WF | −4.783 | *** | −3.387 |
| Ln(EMP) | 1.295 | *** | 12.948 |
| Intercept | −7.022 | ** | −2.286 |
| Industry dummies | Included | | |
| Year dummies | Included | | |
| N | 1234 | | |
| Adj. $R^2$ | 0.440 | | |
| F-value | 49.483 *** | | |

[1] Refer to Table 1 for the definitions of variables. t-values are in parentheses. *, **, and *** represent statistical significance at the 10%, 5%, and 1% level, respectively.

Table 9 shows the regression results for Equation (2). Model 1 includes only a workplace injury variable (Ln(WDL)), while Model 2 considers the interaction of workplace injuries and CSR performance (Ln(WDL) × CSRD). In Model 1, the coefficient of Ln(WDL) is significantly negative at the 1% level, indicating that workplace injuries have a negative impact on firm value. This result is consistent with Cohn and Wardlaw [16]. They argue that workplace accidents cause substantial direct and indirect costs, thus lowering firm value [16]. Firms bear not only direct costs, such as workers' compensation claim costs, but also indirect costs, such as low productivity and damage to corporate reputation. These injury-related costs increase the company's cash outflow, which negatively affects firm value.

In Model 2, the coefficient of Ln(WDL) is significantly negative, whereas the coefficient of Ln(WDL) × CSRD is significantly positive. These results suggest that a company with many working days lost due to workplace injuries has a low firm value, but this relationship is mitigated for companies with a positive CSR performance. These empirical results support Hypothesis 2. Firm value decreases with workplace injuries [16]; however, the negative impact of workplace injuries on firm value is weakened for companies with a good CSR performance. Because CSR firms consider employees as important stakeholders [13], they will be more proactive in dealing with compensation for and recovery of employees' damages from workplace disasters. They will also be more committed to preventing incidents. Thus, the negative effect of workplace injuries on the firm's future cash flows will be smaller in companies with a good CSR performance. Our results support these arguments.

**Table 9.** The effect of CSR on the relationship between workplace injuries and firm value.

| Variables [1] | Model 1 | | | Model 2 | | |
|---|---|---|---|---|---|---|
| | Coefficients | | t-Value | Coefficients | | t-Value |
| Ln(WDL) | **−0.034** | *** | −3.97 | −0.352 | ** | −2.22 |
| CSRD | | | | −0.042 | | −1.59 |
| Ln(WDL) × CSRD | | | | 0.056 | ** | 2.01 |
| LEV | −0.878 | *** | −19.82 | −0.879 | *** | −19.68 |
| CASH | 348.148 | *** | 4.79 | 343.867 | *** | 4.74 |
| SIZE | 0.009 | *** | 3.40 | 0.010 | ** | 2.28 |
| TAR | −0.044 | * | −1.69 | −0.037 | | −1.42 |
| CFO | 0.496 | *** | 6.64 | 0.501 | *** | 6.70 |
| DIV | 5.788 | *** | 10.70 | 5.769 | *** | 10.52 |
| ATO | −0.126 | *** | −15.25 | −0.124 | *** | −14.85 |
| Intercept | 0.428 | *** | 4.60 | 0.661 | *** | 4.31 |
| Industry dummies | Included | | | Included | | |
| Year dummies | Included | | | Included | | |
| N | 1234 | | | 1234 | | |
| Adj. $R^2$ | 0.507 | | | 0.509 | | |
| F-value | 93.43 *** | | | 84.01 *** | | |

[1] Refer to Table 2 for the definitions of variables. *, **, and *** represent statistical significance at the 10%, 5%, and 1% level, respectively. We correct for heteroskedasticity and autocorrelation using the Newey–West HAC robust variance-covariance estimator.

The results for the other control variables are consistent with previous studies [16]. Debt ratio (LEV) is negatively related to firm value. High debt ratios have a negative effect on firm value because they increase the likelihood of bankruptcy. The coefficients on SIZE, CASH, CFO, and DIV are significantly positive. These results are also in line with Cohn and Wardlaw [16]. They argue that a firm's size, cash holdings, operating cash flows, and a propensity to pay dividends are inverse proxies for how financially constrained the firm is. The positive coefficients of these variables suggest that a financially sound firm has a high firm value. The Adj. R2 of the model is relatively high at 50.0%.

*4.3. Additional Test*

In this study, we measure workplace accidents as the number of working days lost caused by workplace injuries. This measure has the advantage of being able to reflect the severity of the workplace injuries but has the limitation that it is difficult to accurately control the impact of the number of employees.

In Tables 10 and 11, we measure workplace accidents not as the total number of working days lost, but as working days lost per employee. This measure divides the sum of working days lost by the number of employees, which provides more precise control over the effect on the model of employee numbers. Table 8 presents the test result of Hypothesis 1 using working days lost per employee as a dependent variable. The coefficient of CSR is negatively significant, in line with the results in Table 8.

Table 11 presents the test result of Hypothesis 2 using working days lost due to workplace injury per employee as the dependent variable. In Table 11, WDLDM is a dummy variable with a value of 1 if the number of working days lost due to workplace injuries per employee is greater than the median of the sample, or a value of 0 otherwise. The coefficient of WDLDM is negatively significant, while that of the interaction between WDLDM and CSR is positively significant, in line with the results in Table 9.

**Table 10.** Test of Hypothesis 1 using an alternative measure of workplace injuries.

| Variables [1] | Coefficients | | t-Value |
|---|---|---|---|
| CSR | −0.186 | *** | (−2.631 |
| SIZE | 0.115 | *** | (8.242) |
| AGE | 0.001 | | 1.087 |
| MCH | 0.438 | *** | (7.031 |
| ROA | 0.194 | | 0.966 |
| LEV | 0.298 | *** | 3.741 |
| WF | −0.626 | *** | (−2.719 |
| Intercept | −2.240 | *** | (−5.742 |
| Industry dummies | Included | | |
| Year dummies | Included | | |
| N | 1234 | | |
| Adj. R$^2$ | 0.191 | | |
| F-value | 16.364 *** | | |

[1] In Table 8, the dependent variable is WDLM, which represents the working days lost due to workplace injury per employee. Refer to Table 1 for the definitions of other variables. *, **, and *** represent statistical significance at the 10%, 5%, and 1% level, respectively.

**Table 11.** Test of Hypothesis 2 using an alternative measure of workplace injuries.

| Variables [1] | Model 1 | | Model 2 | |
|---|---|---|---|---|
| | Coefficients | t-Value | Coefficients | t-Value |
| WDLDM | −0.024 *** | −2.938 | −0.304 ** | (−1.965) |
| CSR | | | −0.037 | (−1.451) |
| WDLDM × CSR | | | 0.048 * | (1.760) |
| LEV | −0.880 *** | −18.538 | −0.879 *** | (−18.548) |
| CASH | 353.781 *** | 4.640 | 349.665 *** | (4.589) |
| SIZE | 0.007 ** | 2.501 | 0.009 ** | (2.124) |
| TAR | −0.046 * | −1.806 | −0.038 | (−1.482) |
| CFO | 0.494 *** | 7.093 | 0.498 *** | (7.163) |
| DIV | 5.774 *** | 10.828 | 5.778 *** | (10.786) |
| ATO | −0.128 *** | −15.433 | −0.125 *** | (−15.074) |
| Intercept | 0.491 *** | 4.340 | 0.648 *** | (4.142) |
| Industry dummies | Included | | Included | |
| Year dummies | Included | | Included | |
| N | 1234 | | | |
| Adj. R$^2$ | 0.496 | | 0.499 | |
| F-value | | | 59.466 *** | |

[1] In Table 9, WDLDM is a dummy variable with a value of 1 if the number of working days lost due to workplace injury per employee is greater than the median of the sample, or a value of 0 otherwise. Refer to Table 2 for the definitions of other variables. *, **, and *** represent statistical significance at the 10%, 5%, and 1% level, respectively.

Overall, the results in Tables 10 and 11 suggest that our findings do not change qualitatively regardless of the measure of workplace injuries.

## 5. Conclusions

In this study, we examine how a firm's CSR activities affect employee safety. Furthermore, we assess whether a firm's CSR activities can mitigate the negative impact of workplace injuries on firm value. We measure a firm's CSR activity as the sum of its environmental, social, and corporate governance scores issued by the Korea Corporate Governance Service. The severity of workplace accidents is measured by the number of working days lost due to workplace injuries. We test our hypotheses using cross-sectional regression models. Our sample comprises non-financial listed companies in Korea from 2012 to 2014.

We find that companies with a positive CSR performance have fewer working days lost due to workplace injuries. In firms that value CSR, an employee is an important internal stakeholder and is essential for sustainable growth [13]. Our findings imply that socially responsible firms invest more in employee safety and, as a result, are less impacted by workplace injuries. We also find that the rate of workplace injuries is negatively related to firm value, but this negative relationship weakens in companies with good CSR performance. Socially responsible companies are active in the prevention of workplace injuries and in recovering from incidents; therefore, investors respond less negatively to the impact of these issues.

Employees' safety concerns can negatively affect long-term corporate growth. This study expands on previous research on corporate sustainability by providing empirical evidence that CSR has the benefit of ensuring employee safety and reducing the social costs of workplace injuries. However, despite these contributions, our study has the following limitations. First, despite the use of a lead-lag regression model, there may still be a problem of reverse causality. Second, we assume that the number of working days lost due to workplace injuries is inversely related to a firm's investment in employee safety. The reality of these assumptions can influence the interpretation of our findings. Finally, workplace injury data are not open to the public, and we analyze injury data only until 2014. These data constraints may affect the generalization of our findings. We expect these problems to be addressed in future studies.

**Author Contributions:** Conceptualization, J.E.K. and E.S.K.; methodology, J.E.K.; software, J.E.K.; validation, E.S.K.; formal analysis, J.E.K.; investigation, E.S.K.; resources, J.E.K.; data curation, J.E.K.; writing—original draft preparation, J.E.K.; writing—review and editing, E.S.K.; visualization, E.S.K.; supervision, J.E.K.; project administration, J.E.K. All authors have read and agreed to the published version of the manuscript.

**Funding:** This research received no external funding.

**Acknowledgments:** We thank the two anonymous reviewers for their constructive suggestions.

**Conflicts of Interest:** The authors declare no conflict of interest.

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
