# Peer review of "Corporate Social Responsibility and Employee Safety: Evidence from Korea"

_sustainability, doi:10.3390/su12072649_

Round 1

Reviewer 1 Report

I read this study very interestingly. Although there are a few questions, it is judged that this paper is in line with the purpose of this journal. Thank you for allowing me to review a good paper.

Reviewer 2 Report

This paper analyzes the relationship between Corporate Social Responsibility and Employee Safety in Korean reality, which is a paper of very high academic value. The authors are using both non-metric factors that are emerging as new areas of interest in accounting. I believe that the research in this paper has policy implications in many ways to come.

This paper uses Korean data to examine whether companies with a high level of social responsibility have fewer working days lost due to workplace injuries. The authors find that CSR activities have the benefit of ensuring a safer working environment for employees. The basic research question is interesting and reasonable. My comments below are organized by the three main areas explored in the paper.

  1. In table 5, the correlations between SIZE and CSR(0.74***), and SIZE and ln(emp) variables (0.40***), are quite high. In such cases, the authors should consider the possibility of multipolarity problems. If the problem is not properly explained, there may be a problem with the adequacy of the research model and the overall reliability of the research results may be adversely affected. The authors need to check the problem of multicollinearity. Specifically, check whether the maximum value of VIF (Variance Inflation Factor) of each regression model exceeds 10.

  1. Your sample period is limited from 2012 to 2014. This is probably because workplace injury data is private data that is not open to the public. You need to further describe the limitations of the study that the sample period is limited to 2014 due to data constraints. I think data reliability is the most important aspect of research. Therefore, when using data that is not officially published, enough explanations and thresholds for understanding should be specified.

  1. Workplace injury data are produced at the workplace level. In order to examine the determinants of an individual company’s injury, the process of summing up injury data at the workplace level by company should be preceded. Due to the difficulty of summing up workplace-level injury data by company, very few previous studies dealing with the determinants of injury of individual companies. You need to highlight this point as a contribution of your study.

Despite the above revisions, the authors' ideas and themes were very fresh. I think it can be of great help especially in understanding the special situation of Korean labor conditions. I hope that the thesis will be revised more appropriately. So I hope the results of this paper will further strengthen the CSR of Korean companies. I hope it can be used as a good source for investors who are interested in investing in Korean companies to look at Korea's labor situation objectively.

Reviewer 3 Report

Dear Author (s)

The paper addressing a good topic but has some issue to be considered like:

  1. abstract: need to be re-written again in an academic format.
  2. Introduction: The paper contribution is mixed with results. contribution of the paper should be addressed clearly so readers know the value of your research to the previous studies. The last part of introduction, they used the word chapter in drafting the paper structure, preferably they use the word section.
  3. As i observed from correlation table, high possibility of having Autocorrelation and Heterosedasticity in your models. For this reason, you need to run diagnostic tests to confirm having a model clear of Autocorrelation and Heterosedasticity. Its should be done for all models.
  4. the results discussion should be enhanced, more justification required.
  5. the results should be connected to the literature review.
  6. policy implication is required.

Reviewer 4 Report

attached

Round 2

Reviewer 2 Report

Paper was revised properly by my review note so  I accept this paper in present form

Thank you for author's efforts

Reviewer 3 Report

Dear Authors

Thank you for responding to the comments.

Good Luck

Reviewer 4 Report

well done. Congrats!